# Functional biodiversity loss along natural $CO_2$ gradients

Nuria Teixidó [1,2,8], Maria Cristina Gambi [1], Valeriano Parravacini [3], Kristy Kroeker [4], Fiorenza Micheli [2,5], Sebastien Villéger [6] & Enric Ballesteros [7]

The effects of environmental change on biodiversity are still poorly understood. In particular, the consequences of shifts in species composition for marine ecosystem function are largely unknown. Here we assess the loss of functional diversity, i.e. the range of species biological traits, in benthic marine communities exposed to ocean acidification (OA) by using natural $CO_2$ vent systems. We found that functional richness is greatly reduced with acidification, and that functional loss is more pronounced than the corresponding decrease in taxonomic diversity. In acidified conditions, most organisms accounted for a few functional entities (i.e. unique combination of functional traits), resulting in low functional redundancy. These results suggest that functional richness is not buffered by functional redundancy under OA, even in highly diverse assemblages, such as rocky benthic communities.

[1] Stazione Zoologica Anton Dohrn, Department of Integrative Marine Ecology, Villa Dohrn-Benthic Ecology Center, Punta San Pietro Ischia, 80077 Naples, Italy. [2] Hopkins Marine Station, Stanford University, Pacific Grove, CA 93950, USA. [3] Ecole Pratique des Hautes Etudes, CRIOBE, USR 3278, PSL-EPHE-CNRS-UPVD, LABEX Corail, University of Perpignan, 66860 Perpignan, France. [4] University of California, Santa Cruz, Santa Cruz, CA 95064, USA. [5] Center for Ocean Solutions, Stanford University, Pacific Grove, CA 93950, USA. [6] MARBEC, Université de Montpellier-Centre National de la Recherche Scientifique-IRD-IFREMER, University of Montpellier, 34095 Montpellier, France. [7] Centre d'Estudis Avançats de Blanes – CSIC, Blanes 17300 Girona, Spain. [8] Present address: Sorbonne Université, CNRS, Laboratoire d'Océanographie de Villefranche, Villefranche-sur-Mer, France. Correspondence and requests for materials should be addressed to N.Tó. (email: nuria.teixido@szn.it)

There is growing concern for how species loss from human-induced environmental change will affect the functioning of ecosystems and, in turn, the services ecosystems provide to humanity[1–3]. Although groundbreaking work has quantified species loss in response to environmental change[4,5] and has defined the role of species diversity in ecosystem function[6–9], our understanding and predictive ability regarding the effects of environmental change on ecosystem function and services are still extremely limited, particularly in marine ecosystems[10]. High functional diversity (i.e., the breadth of species' ecological functions), functional redundancy (i.e., the number of species performing similar functions in the system) and functional vulnerability (i.e., decrease in functional diversity following the loss of species) are critical for sustaining ecosystem function[8,11,12]. Analyses of change in community functional diversity components in response to environmental change have the potential to connect species loss to shifts in ecosystem function.

Here, we use a functional-trait approach to quantify different components of functional diversity of benthic marine assemblages exposed to ocean acidification (OA). OA reflects a suite of changes in seawater chemistry due to the uptake of excess anthropogenic $CO_2$ by the ocean, resulting in a decline in the surface ocean pH, carbonate ion concentration and saturation state of carbonate minerals, while increasing $CO_2$ and bicarbonate ion concentrations[13,14]. This suite of chemical changes is predicted to have profound impacts on a wide range of marine species, but most studies have addressed these effects in the laboratory[15,16]. Natural carbon dioxide vents provide an unparalleled opportunity to assess species and whole-ecosystem responses to a projected global change driver[17–21]. These systems can provide novel insights into how oceans will function under future OA scenarios.

Functional traits are defined as those traits that directly influence an organism's performance[11,22]. Changes in the occurrence of these traits under different environmental conditions can provide insights into potential loss of ecosystem function in response to perturbations[11,23]. Functional traits have proven to be an exceptionally versatile and sensitive approach to assess functional diversity and redundancy in terrestrial communities[24,25], but our understanding of how marine functional diversity and redundancy respond to global environmental change across taxa and ecosystems remains elusive. Community analyses based on functional traits that capture marine species' vulnerability to climate change and OA can improve our understanding and predictive ability of how the structure and function of marine ecosystems is affected under current and future climate change scenarios.

At natural volcanic $CO_2$ vents in Ischia, Italy[17–19,26,27], the conditions in low pH zones are used to represent future climatic conditions with a decrease in surface pH from −0.14 to −0.4 pH units under IPCC Representative Concentration Pathways (RCPs) RCP2.6 and RCP8.5 by 2100 relative to 1870[16,28], whereas conditions in the extreme low pH zones are used to represent more extreme scenarios based on high $CO_2$ emissions or the more distant future by 2500[29]. Although the pH zones can provide some insight into acidification scenarios, they are not perfect proxies. One important distinction involves increases in the variability of seawater pH with decreasing means. Although variability in pH/$pCO_2$ will increase with dissolved inorganic carbon due to the thermodynamics of the carbonate system[30] in the future ocean, it is not possible to disentangle any effects of changes in the mean vs. variability in this system. Thus, the conditions in the pH zones should be considered as pH regimes, with decreases in mean pH coinciding with increases in variability. Previous studies conducted at $CO_2$ vent ecosystems have reported an overall decrease in species richness, biomass,

structural and trophic complexity and increased abundance of erect macroalgae, seagrasses and soft-corals with increasing acidification[17,20,21,31,32] and alteration of competitive and consumer–resource interactions[19,26,33]. To date, however, functional differences between species that 'win' or 'lose' under OA have not been quantified. Applying functional-trait-based approaches[34] to entire benthic marine assemblages at natural $CO_2$ vents can provide new insights into mechanisms of resilience to OA across taxa, as well as functional changes in diverse ecosystems and geographic regions.

Given the importance of functional diversity in marine benthic communities, we sought to investigate patterns of change in functional diversity to changes in seawater carbonate chemistry. Specifically, we sought to answer the following main questions: (1) Is functional diversity buffered by functional redundancy, such that the loss of functional diversity with acidification is less than taxonomic diversity? (2) Which life-history traits are more vulnerable to OA? To answer these questions, we quantified the per cent cover of 72 benthic species (Supplementary Table 1) on rocky reefs in different environmental conditions (ambient pH, low pH and extreme low pH zones) associated with volcanic carbon dioxide vents[17,18]. We characterize the ecology of each species using 15 functional traits describing growth forms, solitary vs. colonial life histories, seasonality, size ranges, feeding characteristics, presence or absence of calcareous skeletons (Supplementary Note 1, Supplementary Table 2), which resulted in 68 unique trait combinations or functional entities (FE) (defined as groups of species sharing the same trait values[35], see Methods). Analyses indicated that functional richness loss (the volume inside the convex hull surrounding the FEs) is more than two times greater than taxonomic loss in extreme pH conditions, and that most organisms account for a few FEs, which suggests that even highly diverse assemblages, such as rocky benthic communities, are not buffered against the loss of functional diversity under OA. This integrative approach provides new insights into how future oceans will function in the face of OA and indicates that functional loss may be more severe than anticipated based on taxonomic analyses.

## Results

**Functional diversity of benthic assemblages.** The total number of benthic species decreases with acidification, from 55 species in the ambient pH zones to 45 in the low pH zones and 22 in the extreme low pH zones, representing 76%, 62%, and 31% of the global pool (Fig. 1). These pH zones are delimited based on spatial variability in $CO_2$ venting intensity: the pH and carbonate system in the ambient pH zones is comparable to current conditions, whereas the low pH zones are most comparable to projections for the acidification of the near-future surface ocean, and the extreme low pH zone regimes are most comparable to more extreme distant-future scenarios (see Methods for a description of the pH zones). Although, the carbonate chemistry in the extreme low pH zones is not predicted to be widely observed in the near future, these conditions provide a conceptual framework for understanding extreme acidification impacts. The pattern of species loss associated with $CO_2$ venting is consistent with previous analyses of species composition and abundance along these pH gradients[17,19]. In addition to a more than twofold decrease in the number of species, there was also a decrease in the number of FEs (unique combinations of traits values) with acidification, from 75% of the total FEs present in ambient pH to 62 and 31% of the total FEs present in the low and extreme low pH zones, respectively (Fig. 1). The most striking result, however, is the sharp reduction in functional richness (defined as the volume of functional space delineated by traits filled by species[35,36]) in the extreme low pH zones, which contained

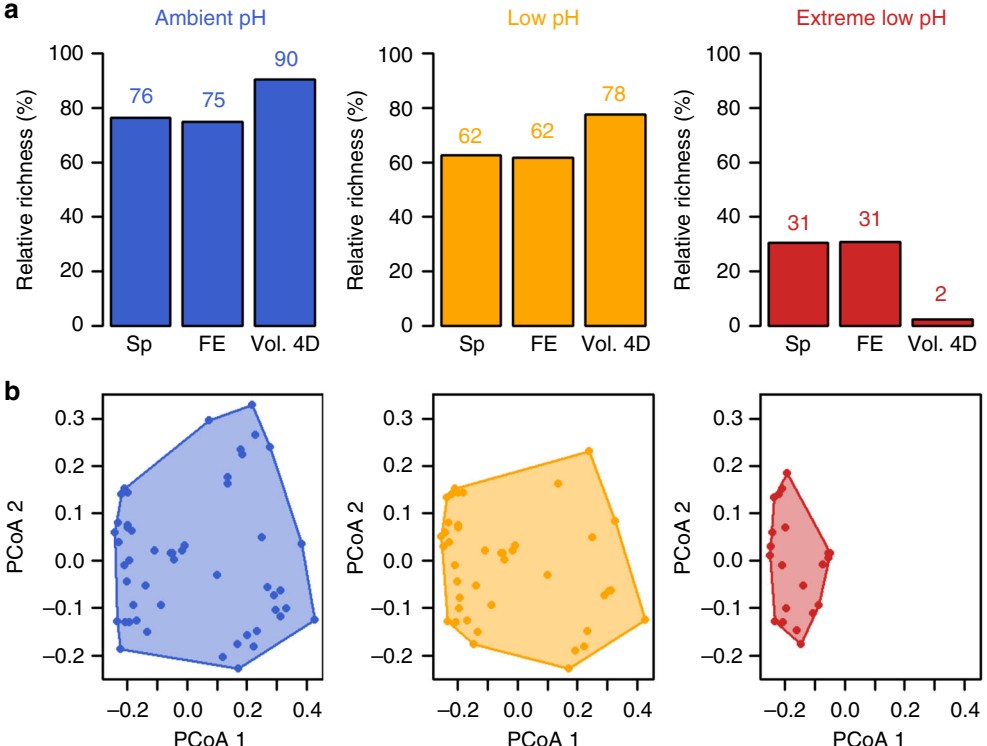

**Fig. 1** Species and functional diversity changes among pH zones. **a** Barplots show species richness (Sp), number of functional entities (unique trait combinations, FE) and functional richness (volume filled by each assemblage in the four dimensions of the functional space, Vol. 4D). Values are expressed as a relative percentage of the value for the total pool and are displayed above the bars. **b** Functional space filled by the functional entities (FEs) present in species assemblages from each pH condition. Axes (PCoA1 and PCoA2) represent the first two dimensions of the 4D functional space. Principal coordinate analysis (PCoA) was computed on functional-trait values. Number of species = 72; number of FEs = 68

only 2% of the global richness, despite the benthic assemblage in this zone containing 31% of species and 31% of FEs present in the global pool.

**Taxonomic and functional β-diversity.** We investigated whether variations of taxonomic β-diversity and of functional β-diversity were mostly driven by turnover (replacement) or nestedness-resultant (richness difference) components using the additive partitioning of dissimilarity (see Methods). Functional β-diversity, measured as the proportion of functional space filled by benthic assemblages that is not shared, was high in ambient-extreme low pH (98%) and low-extreme low pH (97%) (Supplementary Table 3), whereas it was low in ambient-low pH zones (17%) with high overlap in the functional space (Supplementary Figure 1, Supplementary Table 3). Ninety five and 92% of the functional richness of low and extreme low pH were nested in the volume filled by assemblage from ambient pH zone, indicating that the functional-trait values in the acidified zones are a subset of those from the ambient zone (Supplementary Figure 1). In contrast, variation in β-taxonomic diversity was mainly due to species turnover ranging from 0.4 (between low and ambient pH zones) to 0.62 (between extreme low and ambient pH zones) (Supplementary Table 3). These results demonstrate that while the high taxonomic β-diversity is mostly driven by the replacement of species among the pH zones, the high functional β-diversity is driven mainly by a loss of functional richness at low pH. Moreover, we found some conservation of functions between ambient and low pH zones despite the high turnover in taxonomic richness.

**Null models and sensitivity analyses.** We generated a null model to test whether the observed values of functional richness were

significantly different from the null hypothesis that species are randomly distributed into FEs. We found that values of functional richness do not deviate from a random expectation in ambient and low pH zones, but there is a strong environmental filtering of trait values in the extreme low pH zones (Supplementary Figure 2), indicating that the observed values are significantly lower than expected by chance. In the extreme low pH zones, species are more densely packed into a few FEs with similar trait values than one would expect based on a random assignment (Fig. 2, see below). Because functional richness may be affected by the categorization of traits, we also performed a sensitivity analysis to test the robustness of all the results. We reduced the number of FEs to 55 and re-ran all the analyses. We did not find any major changes to the reported results obtained with the finer categorization (Supplementary Figure 3, Supplementary Table 4).

**Loss of FEs along the pH gradient.** The overall loss of functional richness with decreasing pH was coupled with a small decrease in the number of species within FEs, with the mean number of species in each FE declining from 8 in ambient and low pH zones to 6 in extreme low pH zones (Supplementary Figure 4). However, the distribution of abundance among different FEs (another key aspect of functional redundancy) and the vulnerability of FEs (FEs having only one species) changed greatly along the pH gradient (Figs. 2, 3, Supplementary Figure 5). The FEs in ambient pH zones were characterized by a wide variety of morphological forms ranging from encrusting (24%), articulate (13%), to massive-hemispheric forms (3%), with a mixture of seasonal (43%) and perennial (58%) entities that exhibit a range of growth rates (from very high >10 cm/year, 8%, to slow rates 1 cm/year, 11%), a large spectrum of sizes, photosynthetic autotrophy (86%)

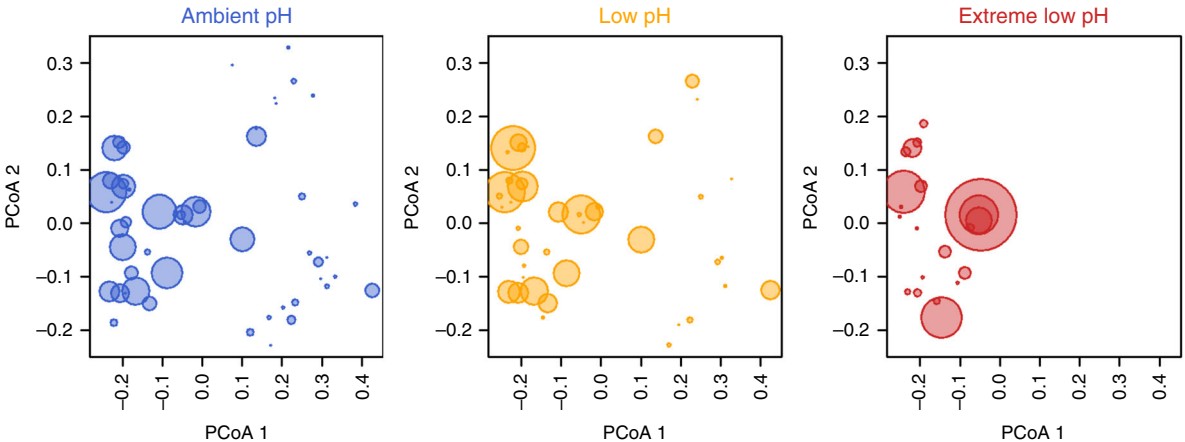

**Fig. 2** Overall distribution of FE abundance across the functional space. Each point represents a functional entity (i.e. unique combination of functional attributes) and the size of the circles is proportional to the relative cover of the species belonging to a certain functional entity. Number of species = 72; number of FEs = 68

and heterotrophy (13%), including filter feeders (14%) and species using grazing/herbivory strategies (<1%), and a wide range of calcification features on their skeletons/bodies (Figs. 2, 3, 4, see Supplementary Table 5 for abundances of functional-trait categories). The FEs in the ambient pH zones describe traits of 'slower' life histories for different morphological forms and sizes of algae and invertebrates of shallow benthic assemblages (Figs. 2, 3, 4, Supplementary Figure 6 and see Table 1 for examples of the species with these traits). Conversely, in the low pH zones, we primarily observed encrusting (25%), filamentous (18%) and foliose erect thallus (14%) morphological forms, with highly seasonal population dynamics (>50%) and high to very high growth rates (60%), medium sizes (from 5 to 50 mm height, 58%), photosynthetic autotrophy (87%), but some heterotrophy (13%) with active feeding strategies (13%), and a lack of calcification (78%), although there were few entities with non-calcareous spicules (3%) and calcified structures (18%) (Figs. 2, 3, 4, Supplementary Table 5). This group of FEs describes traits of mainly fleshy 'seasonal' forms that grow quickly and are mainly composed of non-calcareous algae (Figs. 2, 3, 4, Table 1). Finally, the few, highly redundant FEs in extreme low pH zones were represented by species with encrusting (60%), filamentous (16%) and foliose, erect thallus (14%) morphologies, small sizes (79%) (from 1 to 50 mm in height), mainly seasonal population dynamics (31%), high to very high growth rates (36%) (>10 cm/year), using photosynthesis as the main energy resource (100%), and lacking calcification (99%) and motility (100%) (Figs. 2, 3, 4, Supplementary Table 5). These FEs represent encrusting fleshy forms, 'fast', non-calcareous algal assemblages (Figs. 2, 3, 4, Table 1). Examples of the species with these traits in the extreme pH zones are the algae *Hildenbrandia crouaniorum*, *Dictyota* spp. and algal turf constituents (Table 1, Supplementary Table 1).

## Discussion

Our results show that functional diversity (FE richness, functional richness as the volume surrounding the FEs) decreases with acidification and that functional loss is more pronounced than the corresponding decrease in taxonomic diversity. These results highlight the importance of considering functional diversity and ecological redundancy, in addition to species diversity, for assessing and predicting the effects of OA on marine communities. Our analyses suggest that even moderate changes in species composition can have significant impacts on functional diversity and redundancy, and thus on ecosystem function.

Based on our trait classification, most FEs in these temperate benthic assemblages are represented by just one species (66 FEs out of 68), and thus this ecosystem is highly vulnerable because most of FEs show no functional redundancy or insurance. A similar pattern of low functional redundancy, and thus high vulnerability to impacts from fishing, was previously shown for temperate and tropical reef fish communities[10,37–39]. Moreover, we uncovered an additional mechanism for functional redundancy loss under OA by showing reduced abundance across multiple FEs, with most abundance concentrated in few FEs under extreme low pH conditions. Thus, acidification does not only result in the loss of functional-trait combinations, but also in a redistribution of abundance so that high abundance levels persist only in a small subset of functional-trait combinations. This may further increase the vulnerability of low-abundance FEs to additional stressors from climate change and local anthropogenic impacts, and ultimately may affect the long-term resilience of ecological communities to cope with environmental change[8,35].

Our results reveal which vulnerable FEs are maintained and lost in a naturally acidified ecosystem. In particular, the FE that are preserved under lower and more variable pH include species with encrusting, filamentous and foliose morphologies, seasonal life histories, with high growth rates, photosynthetic autotrophy and lack of calcified structures. Thus, these traits appear to be least vulnerable to OA. Conversely, the FEs that are commonly lost include erect and massive morphologies, long-lived and slow-growing life histories, heterotrophic feeding strategies and calcification.

Reduced abundance, or even loss of entire species with important trait values, can result in the loss of crucial ecosystem processes. For example, loss of long-lived, slow-growing habitat forming species (e.g. macroalgae, corals), which create complex, three-dimensional biological structure have important effects on species diversity, sediment retention, invasion resistance, and resilience to additional disturbance[40,41], with dramatic effects on associated species, ecosystem function and stability[41–43]. Similarly, loss of heterotrophic invertebrates and of calcifying algae and invertebrates may affect carbon storage, nutrient cycling and energy transfer through food webs[18,33,44,45].

Loss of grazers (i.e. limpets and sea urchins) can have cascading effects on the structural and functional robustness of temperate communities because grazers, by feeding on algal biomass, maintain the heterogeneity and patchiness of benthic communities and transfer algal primary production to higher trophic levels[19,33,46,47]. In particular, loss of these intermediate consumers in communities dominated by autotrophs may result

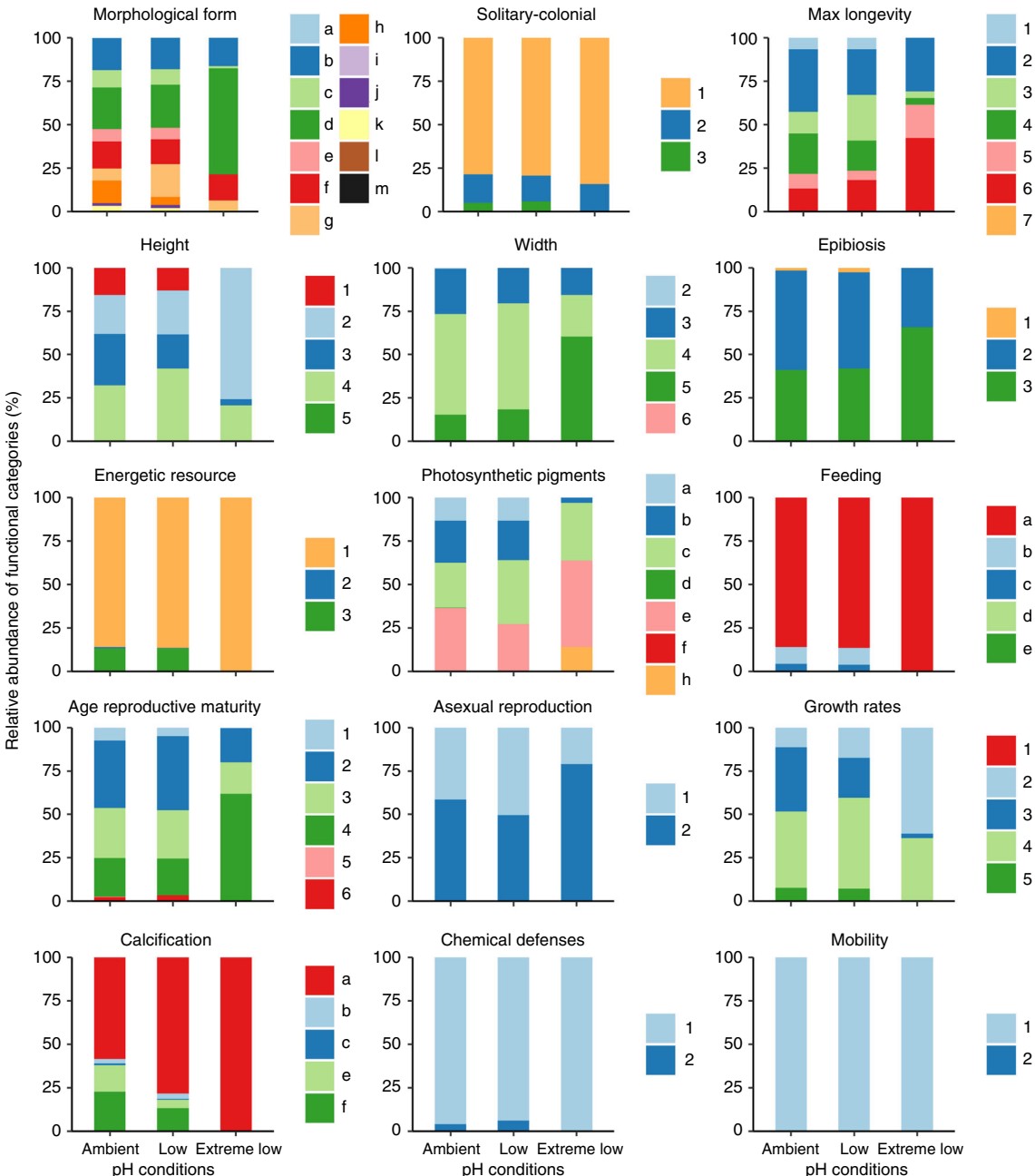

**Fig. 3** Change in relative abundance of functional-trait categories along the pH gradient. Stacked bar graphs show relative abundance of each trait category. Colour scales refer to categories for each trait. Some categories are not visible because their relative abundance is <1 % (see Supplementary Table 5 for abundances). Morphological form: (a) boring, (b) filaments, (c) stolonial, (d) encrusting, (e) encrusting, leaf-like, with blades, (f) foliose erect thallus, sheets/blades, (g) coarsely branched, (h) articulated, (i) cup-like, (j) massive encrusting, (k) massive-hemispheric, (l) massive-erect, (m) tree-like; solitary-colonial: (1) solitary, (2) gregarious, (3) colonial; maximum longevity: (1) weeks, (2) 3–11 months, (3) 1 year, (4) 2 years, (5) 5 years, (6) 10–20 years, (7) >20 years; height: (1) up to 1 mm, (2) 1–10 mm, (3) 10–50 mm, (4) 50–200 mm, (5) 200–500 mm; width: (2) 0.1–1 mm, (3) 1–10 mm, (4) 10–50 mm, (5) 50–200 mm, (6) >200 mm; epibiosi: (1) obligate, (2) facultative, (3) ever; energetic resource: (1) photosynthetic autotroph, (2) photo-heterotroph, (3) heterotroph; photosynthetic pigments: (a) no, (b) Chl a, Chl b, β-carotene, xanthophyll (e.g. green algae), (c) Chl a, xanthophyll /fucoxanthin, Chl c1 + c2 (e.g. brown algae), (d) Chl a, chlorophyll c2, peridinin (e.g. dinoflagellates, present in invertebrates), (e) Chl a, phycocyanin, phycoerythrin (e.g. red algae), (f) Chl a, phycocyanin (cyanobacteria present in sponges), (h) mixture of, (a), (b), (c), (e) (e.g. turf); feeding: (a) no (autotroph), (b) active filter feeders with cilia, (c) active filter feeders by pumping, (d) passive filter feeders, (e) herbivores/grazers; age reproductive maturity: (1) weeks, (2) 3–5 months, (3) 6–11 months, (4) 1 year, (5) 2 years, (6) 2–5 years; asexual reproduction: (1) no, (2) yes; growth rates: (1) extreme slow (<1 cm/year), (2) slow (1 cm/year), (3) moderate (>1 cm/year), (4) high (5–10 cm/year), (5) very high (>10 cm/year); calcification: (a) without, (b) non-calcareous spicules, (c) calcareous spicules and sclerites, (e) carbonate with discontinuities, (f) continuous carbonate; chemical defenses: (1) no, (2) yes; mobility: (1) sessile, (2) vagile. See Supplementary Table 2 for trait category descriptions. Traits, n = 15; trait categories, n = 73

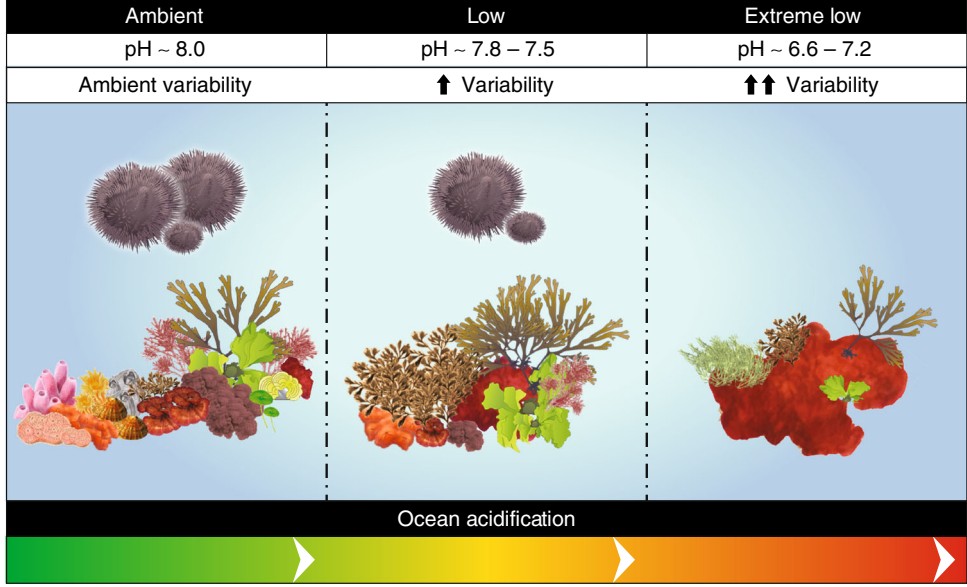

| Ambient | Low | Extreme low |
|---|---|---|
| pH ~ 8.0 | pH ~ 7.8 – 7.5 | pH ~ 6.6 – 7.2 |
| Ambient variability | ⬆ Variability | ⬆⬆ Variability |

Ocean acidification

Biodiversity loss (taxonomic and functional)

**Fig. 4** Taxonomic and functional biodiversity loss along the pH gradient. The ambient pH zone is characterized by a mosaic of strategies, from 'faster' to 'slower' life histories, from encrusting to massive and erect forms, including a variety of sizes, both photosynthetic autotrophy and heterotrophy (including filter feeders and grazers/herbivores), and the presence of calcareous skeletons. The low pH zone is mainly characterized by fleshy morphologies, seasonal population dynamics, fast growth and is mainly composed of non-calcareous organisms, where photosynthetic autotrophy is the major energetic resource. The conditions in low pH zones are used to represent atmospheric carbon dioxide concentration values under future climatic conditions with a decrease in surface pH from −0.14 to −0.4 pH units under IPCC RCP2.6 and RCP8.5 by 2100 relative to 1870. The extreme low pH zone is dominated by encrusting-fleshy forms, 'fast' growth, non-calcareous organisms and photosynthetic autotrophy as the only energetic resource. The encrusting red form is *Hildenbrandia crouaniorum*, a non-calcareous, perennial red algae. This extreme low pH zone is used to represent more extreme scenarios based on high $CO_2$ emissions or the more distant future by 2500. See Table 1 for names of selected species supporting ecological functions

in reduced energy and biomass transferred to high trophic levels, with possible impacts on fish populations and fisheries productivity[37,48]. Dominance by short-lived species may also result in decreased stability, with high variation in productivity and potential impacts on higher trophic levels[46,49,50]. This example further highlights how systematic loss of functionally important species and their interactions can have profound impacts on how ecosystems function under OA.

The study system exhibits high temporal variability in seawater pH due to $CO_2$ venting intensity and subsequent mixing, as well as fundamental thermodynamics in the carbonate system[30]. As a result, as mean pH values decrease across the pH zones, pH variability and extreme pH fluctuations increase. A growing body of experimental evidence suggests that variability in carbonate chemistry can mediate species responses to changes in mean pH/ $pCO_2$[51], but it is not possible to disentangle these two drivers in this system. More research determining how individual species and assemblages respond to temporal variability can provide valuable insights into mechanisms underlying community changes with implications for ecosystem function.

Previous studies from natural $CO_2$ vent systems have provided critical insights regarding the impacts of OA on the structure and complexity of intact benthic communities. Here, we add to these insights by using functional traits to advance our understanding of the emergent ecological consequences of OA for marine communities and link changes in structure to potential effects on ecological function. Functional-trait approaches can be broadly applied across ecosystem types to improve our understanding of how functional diversity affects ecosystem processes. This will be critical to better comprehend the generalities of the vulnerability of marine benthic communities under ongoing and predicted global environmental changes.

## Methods

**Study sites.** The volcanic $CO_2$ vents are located at 0.5–3 m depth on the north and south sides of the islet Castello Aragonese at Ischia Island (Italy; 40 43.840′N, 13 57.080′E) adjacent to sloping rocky reefs. The north and south sites are 150 m apart and separated by a bridge connecting the Castello islet to Ischia. At both sites, each pH zone is ~20 m in length and separated from the next zone by at least 20–25 m[17,19]. Water carbonate chemistry and in situ monitoring of seawater pH delineated a pH gradient with three carbonate chemistry zones (ambient, low and extreme low pH) caused by spatial variability in $CO_2$ venting intensity[17]. Reductions in mean pH in each zone is associated with increased temporal variability in pH[19]. For this study, we followed the delineation of three pH zones at the two sites (north and south), which corresponded with the zones used in previous studies[19,26,27]. The pH values were reported as (mean ± SD): pH = 7.21 ± 0.34 north side, pH = 6.59 ± 0.51 south side in extreme low pH zone (high venting activity); pH = 7.77 ± 0.19 north side, pH = 7.75 ± 0.31 south side in low pH zone (moderate venting activity), pH = 7.95 ± 0.06 north side, pH = 8.06 ± 0.09 south side in ambient pH zone (non-visible vent activity) between 0.5 and 1.5 m depth[19]. The mean carbonate chemistry in the ambient pH zones correspond to current average conditions, whereas the low pH zones are most comparable with values predicted for the year 2100 with a decrease of pH from −0.07 to −0.33 pH units under RCP2.6 and RCP8.5 and extreme low pH zones approach the most extreme acidification scenarios for 2500[26,27]. Because the temporal variability in pH increases as the mean pH decreases, it is not possible to disentangle the possible effects of changes in mean vs. the variability. Therefore, it is important to consider these zones as 'pH regimes', with decreasing mean and increasing variability. Due to the fundamental thermodynamics of the carbonate system, increases in dissolved inorganic carbon will increase the variability in pH and $pCO_2$ in the future as well[30,52]. See refs. [18,19,26] for more information on the study site and pH variability and geochemical parameters.

**Sampling design.** Per cent cover of benthic species on the rocky reef was quantified using visual census techniques in June 2015 and June 2016. Twelve quadrats of 25 cm × 25 cm were haphazardly placed along the rocky reef at 0.5–1.5 m depth at the two sites in each of the three pH zones ($n = 72$ quadrats in total), which corresponded to the depth range sampled for carbonate chemistry and in situ pH measurements. Each quadrat was divided into a grid of 25 squares (5 cm × 5 cm each) and % cover was quantified by counting the number of squares filled in the grid by the species and expressing the final values as relative percentages[53,54]. We

### Table 1 Summary of selected species and their functional traits found among the pH zones

| Taxa | pH zones | | | Common names and traits |
|---|---|---|---|---|
| | **Ambient** | **Low** | **Extreme low** | |
| *Paracentrotus lividus* | ✓ | – | – | Sea urchin, massive, perennial (10–20 years), heterotroph, herbivore/grazer, slow growth rates, calcified |
| *Patella caerulea* | ✓ | – | – | Limpet, cup-like, perennial (5 years), heterotroph, herbivore/grazer, slow growth rates, calcified |
| *Perforatus perforatus* | ✓ | – | – | Barnacle, massive, perennial (5 years), heterotroph, active filter feeder, slow growth rates, calcified |
| *Balanophyllia europaea* | ✓ | – | – | Coral, cup-like, perennial (10–20 years), heterotroph, passive filter feeder, extreme slow growth rates, calcified |
| *Chondrosia reniformis* | ✓ | – | – | Demosponge, massive encrusting, perennial (>20 years), heterotroph, active filter feeder, slow growth rates, non-calcified |
| *Corallina elongata* | ✓ | ✓ | – | Red alga, articulated, perennial (2 years), autotroph, moderate growth rates, calcified |
| *Neogoniolithon brassica-florida* | ✓ | ✓ | – | Red alga, encrusting, perennial (10–20 years), autotroph, moderate growth rates, calcified |
| *Peyssonnelia rosa-marina* | ✓ | ✓ | – | Red alga, encrusting with blades, perennial (5 years), autotroph, slow growth rates, calcified |
| *Peyssonnelia squamaria* | ✓ | ✓ | – | Red alga, encrusting with blades, seasonal (<1 year), autotroph, moderate growth rates, non-calcified |
| *Crambe crambe* | ✓ | ✓ | – | Demosponge, encrusting, perennial (10–20 years), heterotroph, active filter feeder, moderate growth rates, non-calcified |
| *Dictyota* sp. | ✓ | ✓ | ✓ | Brown alga, fleshy erect, seasonal (<1 year), autotroph, high growth rates, non-calcified |
| *Halopteris scoparia* | ✓ | ✓ | ✓ | Brown alga, coarsely branched, perennial (1 year), autotroph, high growth rates, non-calcified |
| *Hildenbrandia crouaniorum* | ✓ | ✓ | ✓ | Red alga, encrusting, perennial (10–20 years), autotroph, slow growth rates, non-calcified |
| *Flabellia petiolata* | ✓ | ✓ | ✓ | Green alga, stolonial form, perennial (2 years), autotroph, high growth rates, non-calcified |
| Turf algae | – | ✓ | ✓ | Mixture of algae, filaments, seasonal (weeks), autotroph, high growth rates, non-calcified |

See Supplementary Table 3 for % cover of species, Supplementary Table 4 for the entire description of traits and their codes and Supplementary Table 5 for abundance of functional-trait categories
Growth rates: Extreme slow (<1 cm/year), slow (1 cm/year), moderate (>1 cm/year) and high (5–10 cm/year)

identified a total of 72 benthic taxa (43 algal species, 1 algal turf species group and 28 invertebrate species), which were identified at the lowest taxonomic level (Supplementary Table 1).

**Functional characterization of benthic species**. The functional diversity of benthic assemblages was assessed using 15 traits, which describe complementary facets of species ecology: morphological form, solitary-colonial life histories, maximum longevity, height, width, epibiosis, energetic resource use, major photosynthetic pigments, feeding, age at reproductive maturity, potential asexual reproduction, growth rates, physical defenses (calcification), chemical defenses and mobility (see Supplementary Note 1 and Supplementary Table 2 for a detailed description of the traits). Trait values for the 72 benthic taxa were obtained from the expertise of the team members, by consulting other experts, and from literature review (See Supplementary Note 1). This study focused on global comparisons of traits across species and not on intraspecific variation. Trait values represent the most accurate average description of species-specific traits (e.g. size) without accounting for potential intraspecific variability[34]. FEs were defined as groups of species sharing the same trait values[35]. In total, 72 benthic taxa were classified into 68 different FEs: 55 species and 51 FEs in ambient, 45 species and 42 FEs in low and 22 species and 21 FEs in extreme low pH zones (see below for the number of FEs calculations).

**Functional diversity of benthic assemblages**. We tested and quantified whether functional diversity was reduced along natural $CO_2$ gradients, and whether functional redundancy buffered expected losses caused by reductions in taxonomic diversity. In this study, we pooled the data at both sides (north and south) together to have a general pattern of the effects of acidification, thus resulting in three pH zones: ambient, low and extreme low pH. Before pooling the data from both sides, we performed a set of preliminary analyses to test for statistical differences between north and south sides by using non-parametric analysis of variance PERMANOVA (permutational multivariate analysis of variance) with side and pH as fixed factors. The null hypothesis of no difference in species and FEs composition was tested using 9999 unrestricted permutations of raw data and Type III SS. These preliminary analyses did not reveal any significant differences when considering side

as a factor (Pseudo-$F = 1.4$, $p > 0.05$ for species; Pseudo-$F = 2.8$, $p > 0.05$ for FEs, Supplementary Figure. 7). Then, for each pH zone, we calculated: (i) the FE richness as the number of FEs present in the assemblage relative to the number from the global pool; and (ii) the functional richness as the volume inside the convex hull surrounding the FEs present[35,36]. The number of species, number of FEs and functional volume filled by each assemblage were expressed as a relative percentage of the global pool. To build a functional space accounting for trait values of FEs, we calculated pairwise functional distances between species pairs based on the 15 functional traits using the Gower metric, which allows mixing different types of variables while giving them equal weight[55]. Then, we used the first four principal axes of the principal coordinates analysis (PCoA) computed on the functional distance matrix to build a multidimensional functional space[11,36,56,57], where the position of FEs represents their differences. We selected the number of axes to build a functional space accordingly to the mean squared-deviation index (mSD) computed between initial functional distance among FEs (i.e. based on trait values) and final Euclidean distance in the functional space[57]. Functional space with four dimensions had a low mSD (0.0035) and showed an optimal ability to represent functional differences between species while keeping computational time reasonable for functional β-diversity analyses.

**Null models for testing differences in functional diversity**. We tested whether the observed values of functional richness were significantly different from the null hypothesis that species are randomly distributed into functional entities. In each of the pH zones, we simulated a random assignment of species to FEs, while ensuring that each FE had at least one species. We simulated 9999 random assemblages for each pH zone, while keeping the number of species and the number of FEs constant. The observed values of functional richness were compared with the simulated values using a bilateral test ($\alpha = 5\%$).

**Functional redundancy**. Functional redundancy and vulnerability were calculated following ref [58]. We tested whether functional redundancy and vulnerability of FEs were reduced along the pH gradient. Functional redundancy is defined as the level of functional equivalence among species in an ecosystem, such that one function may be performed by one or many species, and one species may substitute for

another in the latter case[59]. In this study, redundancy was assessed according to the Ripley's k index. Specifically, for each FE, we computed its redundancy counting the number of species and their relative abundance (expressed as % substrate cover) within fixed radius in the functional space. Here we used $k = 1\%$ of the maximum distance between FEs in the functional space, but different definitions (i.e. 0.5%, 5%) produced similar results. From these data, we produced the distribution of redundancy for each FE expressed as species richness and abundance. We adopted this criterion to include abundance of species within FEs because it represents a complementary aspect of redundancy[37,39]. This approach determines whether abundance is packed into particular functions and which functions are the most vulnerable to reductions in abundance. Vulnerability is related to the lack of insurance provided by functionally similar species. Vulnerability was therefore estimated based on the number of species that share similar combination of traits. Following[58] functional vulnerability was then expressed as the proportion of FE in an assemblage that had redundancy of 1 (i.e. the proportion of FEs having only one species within the k radius in the functional space).

**Taxonomic and functional β-diversity.** We assessed the taxonomic and functional β-diversity among benthic assemblages from the three pH zones using the β-diversity partitioning framework[60,61] based on the Jaccard dissimilarity index[62,63]. β-diversity equals 0 when communities are identical and equals 1 when communities are maximally dissimilar (e.g. no species shared for taxonomic β-– diversity)[63]. Regarding functional β-diversity, it equals zero when the portions of the functional space filled by species assemblages are perfectly overlapping, and equals unity when assemblages do not intersect in that functional space. We then investigated whether taxonomic and functional β-diversity associated to OA was mostly due to turnover (i.e. differences in species or functional-trait values between ambient and acidified zones were due to a replacement) or due to nestedness-resultant processes (i.e. species and functional traits of the acidified zone represented a subset of those found in the ambient zone)[63,64]. Turnover equals 0 when communities are functionally nested and equals 1 when no species are shared by the communities[63]. Taxonomic and functional β-diversity and their respective components were computed following refs. [60,63], respectively.

**Sensitivity analyses.** Functional diversity estimates may be affected by the identity of traits. We tested whether our functional diversity estimates were robust to the resolution of the categorization of functional traits. A coarse categorization could potentially produce low functional sensitivity since FEs would be likely represented by many species, while a fine categorization would lead to the opposite, a high sensitivity with many FEs having only one species. Accordingly, we reduced the number of categories for each trait and re-ran all the analyses. This new coarse categorization presented from two to four categories for each trait (Supplementary Table 4 for the decreased number of categories).

Functional diversity analyses were computed using the R functions from the 'FD'[65], 'tripack'[66], 'geometry'[67], 'matrixStats'[68] and 'betapart'[69] R package (R v 3.4.1, R development Core Team, 2017). Preliminary analyses to test for statistical differences between north and south sides were computed using the program Primer v6 with the PERMANOVA + add-on package.

## Data availability

Data files that support the findings and R codes used to analyse the data and generate the results presented in this study can be obtained from https://doi.org/10.5281/zenodo.1475464[70].

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

## Acknowledgements

This research was supported by the National Geographic Society (Window to the Future, Grant No. 9771-15) and the Total Foundation (High $CO_2$ Seas project, Grant No. BIO-2016-081-4). N.T. was supported by a Maire Curie-Cofund (FP7-PEOPLE-Marie Curie Bandiera-Cofund, GA No. 600407) and by a Marie Sklodowska-Curie Global Fellowship under the European Union's Horizon 2020 research and innovation programme (H2020-MSCA- IF- 2015, GA No. 702628). We thank Capitan V. Rando and P. Sorvino (ANS Diving, Ischia) for their field assistance. We also thank E. Cebrian, S. de Caralt, X. Turon, M.J. Uriz, J. Vacelet for their help with trait definitions and J. Vidaurre for assistance with Fig. 4. N.T. thanks A. Smith for helping with R code. We thank R. Elahi for comments, which greatly improved this manuscript.

## Author contributions

N.T. and V.P. designed the study. E.B., K.K., F.M., M.C.G and N.T. were involved with fieldwork. E.B. identified the species. E.B. and N.T. defined the traits. V.P., N.T. and S.V. analysed the data and produced figures. N.T. drafted the inital manuscript and all authors contributed discussion, writing and interpretation.

## Additional information

**Competing interests:** The authors declare no competing interests.

