## [Peer Review File · Nature Communications]

Reviewers' Comments:

Reviewer #2:

Remarks to the Author:

Overall I thought this was a well-written and interesting contribution that uses the very unique CO₂ vent system off Ischia, Italy to examine potential losses in taxonomic and functional diversity associated with ocean acidification in situ. I found the approach and results interesting and analytical approach robust.

I thought there were some missed opportunities, however, as the approach taken in the manuscript was larger descriptive of patterns of change instead of testing specific hypotheses about the ways in which functional diversity may respond to OA as the approach. For example, the authors examine functional vulnerability of FE's and found many vulnerable FE's in the ambient pH zone. Why not explicitly test whether the vulnerable FE's were in fact the ones lost in low and extreme low pH zones? I suspect this is true, but it is unclear based on the data presented and I don't see anywhere in the supplement or main text where the exact number of vulnerable FE's is presented.

Furthermore, I think the manuscript could be improved by more specific wording in some instances to clarify the results. For example, functional diversity is a general term and may be used to describe multiple facets of biodiversity (e.g., richness, evenness, redundancy) and sometimes used to refer to composite metrics that integrate changes in evenness and richness, for example. I think in most cases the authors really mean richness, so it would be good to state that explicitly (e.g., line 100 – this is not the only instance).

More detailed/minor comments below.

Line 32-34: Isn't this true in all these zones, that most organisms were concentrated in few FE's?

Lines 34-36: And that loss species based on functional traits is non-random and selective?

Line 50: Is this really the correct definition of functional vulnerability?

Line 100: What exactly do you mean by functional diversity loss - do you mean loss in functional richness or number of FEs?

Line 199-200: Report how many in FE's present in ambient zone to be consistent.

Lines 129-130: This wording is unclear – do you mean functional beta diversity is the interaction of volumes of the or the opposite of that (e.g., disjunctive union)? Because high overlap would suggest low beta diversity which I don't think is what you mean.

Lines 132-140: Do these results also suggest that there is some conservation of functions even those species identity turns over quickly between the ambient and low pH zone? There is quite high overlap in functional space between the ambient and low pH zones in spite of the high turnover in taxonomic richness.

Lines 153: Change "functional diversity" to "functional richness"

Lines 154: do you mean "mean number of species in each FE"?

Line 155: Missing mean number of species in low pH zone.

Lines 158-183: Why not compare richness of individual traits or calculate community-weighted mean trait values to better quantify this?

Line 214: Do you mean species diversity of higher trophic levels? Otherwise the logic seems circular – loss of some species leads to lower species diversity.

Line 234: Temper this statement to say “potential effects of ecosystem function” since you don’t measure function explicitly.

Line 392: In the introduction the extreme pH zones was suggested to correspond to a 2500 scenario and 2300 is used here.

Reviewer #3:

Remarks to the Author:

This paper uses benthic cover data along a pH gradient at CO₂ vents as evidence for loss of functional entities in response to ocean acidification. This paper was a pleasure to read. It is well written and uses sophisticated and well thought-out statistical techniques to describe differences in functional traits between ambient, low, and extremely low pH zones. The science is sound, but I have a few issues with the interpretation of the results, specifically with respect to making inference for how ocean acidification will affect functional diversity.

These CO₂-vents provide a great platform for testing how pH affects different ecological processes. However, the low and extremely low pH sites also have a highly sporadic pH environment unlike any “natural” rocky reef site and, other than changes in the mean, are not quite analogous to predicted OA scenarios. I suggest that the authors remove some of the specific connections to predicted OA (for example, lines 112-114, title, figures, etc.) and discuss the results in the context of differences in pH mean and variance. On that same note, this study ignores the fact that both mean and variance in pH are changing along this gradient in concert. Because changes in environmental variability can have different effects on ecosystem functioning than changes in means, I would suggest that the authors discuss how differences in pH variance could have affected their results.

Minor comments listed below:

Line 82 says year 2500, but line 392 says year 2300. There is also a reference to the “extreme” year in the figure legends. Which year is it?

Line 155: I think you are missing a number after “low pH to”

Methods:

When were the surveys done? Is the 0.5 – 3m depth gradient co-linear with pH?

Line 437: change “fourth” to four. What is the reason for using the first 4 axes to calculate volume? The # of axes used should have some statistical meaning. For example, the number of axes that explained 95% of the variance in the data.

List the packages used for each analysis when describing the specific analysis rather than listing all the packages at the end of the methods.

I highly encourage the authors to make their code and data publically available for transparency and reproducibility.

Figure 1 legend: Change "histograms" to barplots. Use the same abbreviations in both the text and the figures. For example, you use "FE" in the text and "NbFE" in the figure as well as "functional richness" vs Vol.4D.

Table 1: What is meant by "key" species? Are these the dominant species?

Figure S2: Error bars or points are off

REVIEWER COMMENTS: line numbers refer to original submission

OUR RESPONSES: line numbers refer to revised word doc with track changes

Reviewers' Comments:

Reviewer #2 (Remarks to the Author)

Overall I thought this was a well-written and interesting contribution that uses the very unique CO₂ vent system off Ischia, Italy to examine potential losses in taxonomic and functional diversity associated with ocean acidification in situ. I found the approach and results interesting and analytical approach robust.

We thank the reviewer for the positive comments.

I thought there were some missed opportunities, however, as the approach taken in the manuscript was larger descriptive of patterns of change instead of testing specific hypotheses about the ways in which functional diversity may response to OA as the approach. For example, the authors examine functional vulnerability of FE's and found many vulnerable FE's in the ambient pH zone. Why not explicitly test whether the vulnerable FE's were in fact the ones lost in low and extreme low pH zones? I suspect this is true, but it is unclear based on the data presented and I don't see anywhere in the supplement or main text where the exact number of vulnerable FE's is presented.

We have re-written the objectives in the introduction to emphasize the original questions of the manuscript (see lines 99-106). We have also revised the discussion (see lines 231-233) and methods (see lines 332-334, 371-372) to explicitly emphasize our statistical tests. In addition, we provide the number of vulnerable FE's among pH zones as a new figure in supplementary materials (Figure S5 Vulnerability of FEs among pH zones) and presented it in the results (lines 179-181).

Lines 99-106: “Given the importance of functional diversity in marine benthic communities, we sought to investigate patterns of change in functional diversity to changes in seawater carbonate chemistry. Specifically, we sought to answer the following main questions: 1) Is functional diversity buffered by functional redundancy, such that the loss of functional diversity with acidification is less than taxonomic diversity? 2) Which life-history traits are more vulnerable to ocean acidification? To answer these questions, we quantified the percent cover of 72 benthic species (Supplementary Table 1) ...

Lines 231-233 (Discussion): Our results reveal which vulnerable FEs are maintained and lost in a naturally acidified ecosystem. In particular, the FE that are preserved under lower and more variable pH include ...

Lines 332-334 (Methods): We tested and quantified whether functional diversity was reduced along natural CO₂ gradients, and whether functional redundancy buffered expected losses caused by reductions in taxonomic diversity.

Lines 371-372 (Methods): Functional redundancy and vulnerability were calculated following⁵⁵. We tested whether functional redundancy and vulnerability of FEs were reduced along the pH gradient.

Lines 179-181 However, the distribution of abundance among different FEs (another key aspect of functional redundancy) and the vulnerability of FEs (FEs having only one species) changed greatly along the pH gradient (Fig. 2, Supplementary Figs. S4, S5)

New Figure S5. Vulnerability of FEs among the pH zones. Vulnerability is expressed as the proportion of FE in an assemblage that had redundancy of 1 (i.e. the proportion of FEs having only one species within the k radius in the functional space). We characterized the traits of 72 benthic species, which resulted in 68 unique trait combinations or functional entities (FE). Accordingly, 66 FEs were classified as vulnerable with having only one species: n=49 Ambient pH, n= 40 Low pH, n= 20 Extreme low pH.

Furthermore, I think the manuscript could be improved by more specific wording in some instances to clarify the results. For example, functional diversity is a general term and may be used to describe multiple facets of biodiversity (e.g., richness, evenness, redundancy) and sometimes used to refer to composite metrics that integrate changes in evenness and richness, for example. I think in most cases the authors really mean richness, so it would be good to state that explicitly (e.g., line 100 – this is not the only instance).

We have revisited the use of the general term functional diversity (including multiple facets of biodiversity) and the more specific terms (functional richness, functional redundancy) throughout the manuscript.

Lines 31-32: We found that functional diversity is greatly reduced with acidification, and that functional richness loss is more pronounced than the corresponding decrease in taxonomic diversity.

Lines 35-36: These results suggest that functional richness is not buffered by functional redundancy under OA, even in highly diverse assemblages, such as rocky benthic communities.

Lines 114-115: Analyses indicated that functional richness loss is more than two times greater than taxonomic loss in extreme pH conditions, and that most organisms account for a few functional entities, which suggests that even highly diverse assemblages, such as rocky benthic communities, are not buffered against the loss of functional diversity under OA.

Lines 170-171: Because functional richness may be affected by the categorization of traits, we also performed a sensitivity analysis to test the robustness of all the results.

Line 175: The overall loss of functional richness with decreasing pH was coupled with a small decrease in the number of species within FEs,

Lines 208-209: Our results show that functional diversity (FE richness, functional richness as the volume surrounding the FEs) decreases with acidification and that functional loss is more pronounced than the corresponding decrease in taxonomic diversity.

More detailed/minor comments below.

Line 32-34: Isn't this true in all these zones, that most organisms were concentrated in few FE's?

Yes. This is right. We already mentioned this low redundancy in the results and discussion sections (see lines 175 and 217). However, we could not extend this point in the abstract due to space limitation.

Lines 175-178: The overall loss of functional diversity with decreasing pH was coupled with a small decrease in the number of species within FEs, with the mean number of species per FE declining from 8 in ambient and low pH to 6 in extreme low pH zones (Supplementary Fig. S4).

Lines 217-219: Based on our trait classification, most FEs in these temperate benthic assemblages are represented by just one species (66 FEs out of 68), and thus this ecosystem is highly vulnerable because most of FEs show no functional redundancy or insurance.

Lines 34-36: And that loss species based on functional traits is non-random and selective?

We found that the loss of functional richness was random in ambient and low pH zones but not in extreme low pH zone. These results are mentioned in Lines 165- 168 and in Figure S2: Null model of functional richness (functional volume) among pH zones. However, we could not extend this point in the abstract due to space limitation.

Lines 165- 168: We found that values of functional richness do not deviate from a random expectation in ambient and low pH zones, but there is a strong environmental filtering of trait values in the extreme low pH zones (Supplementary Fig S2), indicating that the observed values are significantly lower than expected by chance.

Line 50: Is this really the correct definition of functional vulnerability?

In the introduction, we mentioned that functional diversity, redundancy and vulnerability are critical for sustaining ecosystem function. These three key concepts were briefly defined to guide the reader (Lines 49-52). We agree with the referee that the definition that we originally used in

the introduction “functional vulnerability (i.e. the decrease in functional diversity following the loss of species)” is a concept that would needs more wording. We expanded the definition of functional vulnerability in Methods in the revised version.

Lines 385– 387: Vulnerability is related to the lack of insurance provided by functionally similar species. Vulnerability is therefore estimated based on the number of species that share similar combination of traits.

Line 100: What exactly to do you mean by functional diversity loss - do you mean loss in functional richness or number of FEs?

We meant “loss in functional richness” as the volume inside the convex hull surrounding the FEs present”. We have re-written this sentence.

Lines 114 – 115: Analyses indicated that functional richness loss (the volume inside the convex hull surrounding the FEs) is more than two times greater than taxonomic loss in extreme pH conditions...

Line 119-120: Report how many in FE’s present in ambient zone to be consistent.

We included the % of FE’s present in the ambient zone in the revised version of the manuscript (lines 136-137). We also added the number of FE’s to clarify the number of species and FEs found in the method section (lines 328-330).

Lines 136-137: In addition to a more than two-fold decrease in the number of species, there was also a decrease in the number of functional entities (FEs, unique combinations of traits values) with acidification, from 75% of the total FEs present in ambient to 62% and 31% in the low and extreme low pH zones, respectively (Fig. 1).

Lines 328-330: In total, 72 benthic taxa were classified into 68 different FEs, with 55 species and 51 FEs in ambient, 45 species and 42 FEs in low, and 22 species and 21 FEs in extreme low pH zones (see below for number of FEs calculations).

Lines 129-130: This wording is unclear – do you mean functional beta diversity is the interaction of volumes of the or the opposite of that (e.g., disjunctive union)? Because high overlap would suggest low beta diversity which I don't think is what you mean.

We modified the text, methods and the legend Figure S1 to make it clearer for this point and the next one (conservation of function between low pH-ambient pH, Lines 160-162), which are related.

Lines 396 - 398 (*Methods: Taxonomic and functional β -diversity*): Regarding functional β -diversity, it equals zero when the portions of the functional space filled by species assemblages are perfectly overlapping, and equals unity when assemblages do not intersect in that functional space. We then investigated whether taxonomic and functional β -diversity associated to OA was mostly due to turnover

Lines 145-149 (Results). Functional β -diversity, defined as the proportion of functional space filled by benthic assemblages that is not shared, was high in ambient pH - extreme low pH (98%) and low pH - extreme low pH (97%) (Table S3), whereas it was low in ambient pH - low pH zones (17%) with high overlap in the functional space (Supplementary Fig. S1, Table S3).

Lines 160-162 (Results). Moreover, we found some conservation of functions between ambient and low pH zones despite the high turnover in taxonomic richness.

Figure S1. Intersection of the three functional volumes among pH zones. Functional β -diversity equals zero when the portions of the functional space filled by species assemblages are perfectly overlapping, and equals unity when assemblages do not intersect in that functional space. Functional β -diversity was high in ambient - extreme low pH (98%) and low - extreme low pH (97%) (Table S3), whereas it showed low values in ambient and low pH zones (17%) with high overlap in the functional space and conservation of functions. In addition, 95% and 92% of the functional volume of low and extreme low pH zones are nested within the volume of ambient pH zones, respectively. 97% of the volume of extreme low pH zone is also nested within the volume of low pH zones. Overall, this means that the trait values of species occurring in acidified

conditions represent a subset of trait values found in the ambient pH zones, indicating environmental filtering caused by decreasing pH.

Lines 132-140: Do these results also suggest that there is some conservation of functions even those species identity turns over quickly between the ambient and low pH zone? There is quite high overlap in functional space between the ambient and low pH zones in spite of the high turnover in taxonomic richness.

Yes. We added some text to clarify this point in the revised manuscript. Please, see the above point, in which we modified the text and the legend Figure S1.

Lines 153: Change “functional diversity” to “functional richness”

Yes. We changed it accordingly to the general point mentioned above.

Lines 175-176: The overall loss of functional richness with decreasing pH was coupled with a small decrease in the number of species within FEs

Lines 154: do you mean “mean number of species in each FE”?

Yes. We corrected it. Thanks. Lines 177-178

Line 155: Missing mean number of species in low pH zone.

Sorry. This was a typographical error. We included the mean number of species in extreme low pH zone. Line 178.

Lines 158-183: Why not compare richness of individual traits or calculate community-weighted mean trait values to better quantify this?

Following the reviewer’s suggestion, we have calculated the relative abundance of functional trait categories and how they changed among the pH zones. This new data emphasizes the distribution of abundances of FEs and their categories along the pH gradient. This is shown in the new Figure S7 in Supplementary materials and addressed in the main text (lines 187 – 206).

New Figure S7. Relative abundance of functional trait categories among pH zones.

Line 214: Do you mean species diversity of higher trophic levels? Otherwise the logic seems circular – loss of some species leads to lower species diversity.

For loss of long-lived, slow, growing habitat-forming species, we meant those species that provide habitat for other species through their creation of three dimensional biological structure, often called foundation species. Some examples of habitat-forming species are seagrasses, macroalgae, corals, gorgonians, and mussel beds. Many of these habitat-forming species are not higher trophic levels. To address this comment, we have added some examples in the Discussion to clarify that we are referring to habitat-forming species.

Lines 241-242: For example, loss of long-lived, slow-growing habitat forming species (e.g. macroalgae, corals), which create complex, three-dimensional biological structure have important effects ...

Line 234: Temper this statement to say “potential effects of ecosystem function” since you don’t measure function explicitly.

We have revised this line and modified.

Line 268-271: Here, we add to these insights by using functional traits to advance our understanding of the emergent ecological consequences of OA for marine communities and link changes in structure to potential effects on ecological function

Line 392: In the introduction the extreme pH zones was suggested to correspond to a 2500 scenario and 2300 is used here.

Sorry. This was a typographical error. The year 2500 is the correct one. Line 297.

Reviewer #3 (Remarks to the Author):

This paper uses benthic cover data along a pH gradient at CO₂ vents as evidence for loss of functional entities in response to ocean acidification. This paper was a pleasure to read. It is well written and uses sophisticated and well thought-out statistical techniques to describe differences in functional traits between ambient, low, and extremely low pH zones. The science is sound, but I have a few issues with the interpretation of the results, specifically with respect to making inference for how ocean acidification will affect functional diversity.

We thank the reviewer for the positive comments.

These CO₂-vents provide a great platform for testing how pH affects different ecological processes. However, the low and extremely low pH sites also have a highly sporadic pH environment unlike any “natural” rocky reef site and, other than changes in the mean, are not quite analogous to predicted OA scenarios. I suggest that the authors remove some of the specific connections to predicted OA (for example, lines 112-114, title, figures, etc.) and discuss the results in the context of differences in pH mean and variance. On that same note, this study ignores the fact that both mean and variance in pH are changing along this gradient in concert. Because changes in environmental variability can have different effects on ecosystem functioning than changes in means, I would suggest that the authors discuss how differences in pH variance could have affected their results.

We agree with the reviewer on the unknown and potential importance of environmental variability on the ecological patterns in natural CO₂ vent systems. Increasing experimental evidence in laboratory studies suggests that temporal variability in carbonate chemistry is important for ecological responses, and understanding how the temporal variability of high or low pH conditions in our study system affects community responses remains a critical area of research. We have revised our language throughout the manuscript to consider this unknown, including following additions in the discussion, and study site description, and Figure 3. Moreover, we modified the text to moderate the statement regarding “predicted OA scenarios”. Finally, we also changed the original title of the manuscript.

Lines 78 - 90: At natural volcanic CO₂ vents in Ischia, Italy^{18-20,27,28}, the conditions in low pH zone are used to represent future climatic conditions with a decrease in surface pH from -0.14 to -0.4 pH units under IPCC Representative Concentration Pathways (RCP) RCP 2.6 and RCP 8.5 by 2100 relative to 1870^{17,29}, whereas conditions in the extreme low pH zone are used to represent more extreme scenarios based on high CO₂ emissions or the more distant future by 2500³⁰. Although the pH zones can provide some insight into acidification scenarios, they are not perfect proxies. One important distinction involves increases in the variability of seawater pH with decreasing means. Although variability in pH/pCO₂ will increase with DIC due to the thermodynamics of the carbonate system³⁰ in the future ocean, it is not possible to disentangle any effects of changes in the mean versus variability in this system. Thus, the conditions in the pH zones should be considered as pH regimes, with decreases in mean pH coinciding with increases in variability.

Lines 127-131 (Results): The pH and carbonate system in the ambient pH zones is comparable to current conditions, whereas the low pH zones are most comparable to projections for the acidification of the near future surface ocean, and the extreme low pH zone regimes are most comparable to more extreme distant-future scenarios (see Methods for a description of the pH zones). The carbonate chemistry in the extreme low pH zones is not predicted in the near future, but provides an endmember scenario for understanding acidification impacts.

Lines 258-266: The study system exhibits high temporal variability in seawater pH due to CO₂ venting intensity and subsequent mixing, as well as fundamental thermodynamics in the carbonate system³⁰. As a result, as mean pH values decrease across the pH zones, pH variability and extreme pH fluctuations increase. A growing body of experimental evidence suggests that variability in carbonate chemistry can mediate species responses to changes in mean pH/pCO₂⁵¹, but it is not possible to disentangle these two drivers in this system. More research determining how individual species and assemblages respond to temporal variability can provide valuable insights into mechanisms underlying community changes with implications for ecosystem function.

Lines 282-303 (Study site): Water carbonate chemistry and *in situ* monitoring of seawater pH delineated a pH gradient with three carbonate chemistry zones (ambient, low and extreme low pH zones) caused by spatial variability in CO₂ venting intensity¹⁸. Reductions in mean pH in each zone is associated with increased temporal variability in pH¹⁹. For this study, we followed the delineation of three pH zones at the two sites (north and south), which corresponded with the zones used in previous studies^{20,27,28}. The pH values were reported as: pH = 7.21 ± 0.34 north side, pH = 6.59 ± 0.51 south side in extreme low pH zone (high venting activity); pH = 7.77 ± 0.19 north side, pH = 7.75 ± 0.31 south side in low pH zone (moderate venting activity), pH = 7.95 ± 0.06 north side, pH = 8.06 ± 0.09 south side in ambient pH zone (non-visible vent activity)²⁰. The mean carbonate chemistry in the ambient pH zones correspond to current average conditions, whereas the low pH zones are most comparable with values predicted for the year 2100 with a decrease of pH from -0.07 to -0.33 pH units under RCP2.6 and RCP8.5 and extreme low pH zones approach the most extreme acidification scenarios for 2500^{26,27}. Because the temporal variability in pH increases as the mean pH decreases, it is not possible to disentangle the possible effects of changes in mean versus the variability. Therefore, it is important to consider these zones as “pH regimes”, with decreasing mean and increasing variability. Due to the fundamental thermodynamics of the carbonate system, increases in dissolved inorganic carbon will increase the variability in pH and pCO₂ in the future as well^{30,52}. See^{19, 20,27} for more information on the study site and pH variability and geochemical parameters.

Title: We have change the previous title “Functional biodiversity loss from ocean acidification” to this new one: “Functional biodiversity loss along natural CO₂ gradients”.

Minor comments listed below:

Line 82 says year 2500, but line 392 says year 2300. There is also a reference to the “extreme” year in the figure legends. Which year is it?

The year 2500 is the correct one. Line 297

Line 155: I think you are missing a number after “low pH to”

Sorry. This was a typographical error. We included the number.

Line 177: with the number of species in each FE declining from 8 in ambient and low pH to 6 in extreme low pH zones (Supplementary Fig. S4).

Methods:

When were the surveys done? Is the 0.5 – 3m depth gradient co-linear with pH?

The field surveys were done in June 2015 and 2016. The depth gradient is not co-linear with pH. Carbonate chemistry and in situ pH data were acquired at 0.5 – 1.5 m depth in all pH zones. The benthic field surveys were carried out at this depth range as well. We have made the following additions in the study site and sampling design sections to clarify this point.

Line 288-292 (Study sites). pH values were reported as: pH = 7.21 ± 0.34 north side, pH = 6.59 ± 0.51 south side in extreme low pH zone (high venting activity); pH = 7.77 ± 0.19 north side, pH = 7.75 ± 0.31 south side in low pH zone (moderate venting activity), pH = 7.95 ± 0.06 north side, pH = 8.06 ± 0.09 south side in ambient pH zone (non-visible vent activity) between 0.5-1.5 m depth²⁰.

Lines 304 – 308. Sampling design: Percent cover of benthic species on the rocky reef was quantified using visual census techniques in June 2015 and June 2016. 12 quadrats of 25 cm x 25 cm were haphazardly placed along the rocky reef at 0.5-1.5 m depth at the two sites in each of the three pH zones (n=72 quadrats in total), which corresponded to the depth range sampled for carbonate chemistry and *in situ* pH measurements.

Line 437: change “fourth” to four. What is the reason for using the first 4 axes to calculate volume? The # of axes used should have some statistical meaning. For example, the number of axes that explained 95% of the variance in the data.

We have corrected “fourth” to four and integrated the reason of using the first 4 axes.

Lines 354-360 (Methods): Then, we used the first four principal axes of the Principal Coordinates Analysis (PCoA) computed on the functional distance matrix to build a multidimensional functional space^{12,36,53,54}, where the position of FEs represents their differences. We selected the number of axes to build a functional space accordingly to the mean squared-deviation index (mSD) computed between initial functional distance among FEs (i.e. based on trait values) and final Euclidean distance in the functional space⁵⁷. Functional space with four dimensions had a low mSD (0.0035) and showed an optimal ability to represent functional differences between species while keeping computational time reasonable for functional beta-diversity analyses.

List the packages used for each analysis when describing the specific analysis rather than listing all the packages at the end of the methods.

The packages 'FD', 'tripack', 'geometry' and 'matrixStats' are necessary for all the analyses. Only the package 'betapart' is specific for the analyses of beta-diversity. Then, we think that it is simpler and clearer to mention all the packages at the end of the methods, to avoid repetition. In addition, all the packages needed are also included in the scripts accordingly, to guide the reader and user.

I highly encourage the authors to make their code and data publically available for transparency and reproducibility.

Yes, data and codes will be posted on a public data repository. Following the data availability rules from Nature Communications, we will include a data availability and code statement immediately upon (approval for) publication.

Lines 419-421: Data and Code availability:

Cover data that support the findings and R codes used to analyze the data and generate the results presented in this study can be obtained from XXX.

Figure 1 legend: Change “histograms” to barplots. Use the same abbreviations in both the text and the figures. For example, you use “FE” in the text and “NbFE” in the figure as well as “functional richness” vs Vol.4D.

We have changed the abbreviations in the figures (Figure 1 and Figures S3 and S8 in supplementary material) to unify with the text in the main manuscript. Due to visualization constraints, we have kept the abbreviation of Vol. 4D in the Figure.

Table 1: What is meant by “key” species? Are these the dominant species?
Key species are those species playing important ecological roles in the community. We changed this term in Table 1 for: “Summary of selected species and their functional traits found among the pH zones” (line 646).

Figure S2: Error bars or points are off

We think that the referee refers that the mean of the observed value of functional richness (point) is not inside the 95% confidence interval (bar) of expected values under a null model for the extreme low pH zone. This is not an error. This is because functional richness is much smaller than expected by chances in extreme low pH zone. We modified the figure legend to clarify this point.

Lines 70-71: Figure S2 legend (Supplementary material). However, for the extreme low pH zone, functional richness is much smaller than expected by chances as the mean observed value is lower than the confidence interval of expected values, indicating a drastic selection of a limited combination of traits (i.e. environmental filtering).

Reviewers' Comments:

Reviewer #3:

Remarks to the Author:

I think the authors did a great job at addressing the comments. I have no further comments on this MS.

Response to reviewers

Reviewer 3 agrees that the figure you added (the number of FEs at each site) is useful, but it does not address the comment about which functions are lost in the transition among CO₂ levels. We therefore request an analysis that explores this loss of functions in more detail, and/or a further elaboration of these changes in the Results section.

We have carefully revised this comment and re-examined all of the analysis performed during the first revision of the manuscript. Supplementary Figure 5 shows the proportion of FEs having only one species (vulnerability of FEs) but Supplementary Figure 7 precisely shows the loss of the relative abundances of functional trait categories and reflects their change along the pH gradient. Specifically, it shows the % change of all the categories analyzed for each of the 15 functional traits. This Figure is the basis of all the conceptual analysis showed in Figure 1 (species and functional diversity) and Figure 2 (distribution of FE abundance). Therefore, following the reviewers' and the Editorial's suggestion, we have decided to include this Figure in the main text as Figure 3 in this final revision, with a detailed figure legend describing the colour scales of each trait category. In addition, we included the data accompanying the loss of abundance of the functional categories in the main text (Results section, subsection "Loss of functional entities (FEs) along the pH gradient", lines 175- 210) and also it is presented as a new Table in the Supplementary Information section (Supplementary Table 5).

Reviewer #3 (Remarks to the Author):

I think the authors did a great job at addressing the comments. I have no further comments on this MS.

We thank the reviewer for helping us to improve the manuscript through her/his comments